# Diet-induced glial insulin resistance impairs the clearance of neuronal debris in *Drosophila* brain

**Mroj Alassaf, Akhila Rajan**◉*

Basic Sciences Division, Fred Hutch, Seattle, Washington, United States of America

* akhila@fredhutch.org

## Abstract

Obesity significantly increases the risk of developing neurodegenerative disorders, yet the precise mechanisms underlying this connection remain unclear. Defects in glial phagocytic function are a key feature of neurodegenerative disorders, as delayed clearance of neuronal debris can result in inflammation, neuronal death, and poor nervous system recovery. Mounting evidence indicates that glial function can affect feeding behavior, weight, and systemic metabolism, suggesting that diet may play a role in regulating glial function. While it is appreciated that glial cells are insulin sensitive, whether obesogenic diets can induce glial insulin resistance and thereby impair glial phagocytic function remains unknown. Here, using a *Drosophila* model, we show that a chronic obesogenic diet induces glial insulin resistance and impairs the clearance of neuronal debris. Specifically, obesogenic diet exposure down-regulates the basal and injury-induced expression of the glia-associated phagocytic receptor, Draper. Constitutive activation of systemic insulin release from *Drosophila* insulin-producing cells (IPCs) mimics the effect of diet-induced obesity on glial Draper expression. In contrast, genetically attenuating systemic insulin release from the IPCs rescues diet-induced glial insulin resistance and Draper expression. Significantly, we show that genetically stimulating phosphoinositide 3-kinase (Pi3k), a downstream effector of insulin receptor (IR) signaling, rescues high-sugar diet (HSD)-induced glial defects. Hence, we establish that obesogenic diets impair glial phagocytic function and delays the clearance of neuronal debris.

## Introduction

Obesity significantly increases the risk for developing neurodegenerative disorders [1–3], yet the precise mechanisms underlying this connection remain unclear. Overconsumption of high-sugar foods is the leading cause of obesity and its comorbidities including type 2 diabetes [4]. The effects of obesity on organ function are wide-ranging, including improper lipid accumulation in non-adipose tissues such as the muscle, cardiac dysfunction, and reduced life expectancy [5–7]. These effects are largely mediated by the breakdown of the insulin signaling pathway. Upon chronic exposure to a high-sugar diet (HSD), circulating insulin levels rise,

**Data Availability Statement:** All relevant data are within the paper and its Supporting Information files.

**Funding:** This work is possible due to grants awarded to AR from the National Institute of

General Medical Sciences (R35GM124593), the Brain Research foundation (BRFSG-2022-09), and the 2023 McKnight Foundation Neurobiology Disorders Award. MA is supported by a postdoctoral fellowship from the Helen Hay Whitney Foundation. The funders had no role in study design, data collection and analysis, decision to publish, or preparation of the manuscript.

**Competing interests:** The authors have declared that no competing interests exist.

**Abbreviations:** Dilp5, *Drosophila* insulin-like peptide 5; ECAR, extracellular acidification rate; G6P, glucose-6-phosphatase; HFD, high-fat diet; HSD, high-sugar diet; IPC, insulin-producing cell; IR, insulin receptor; Ldh, lactate dehydrogenase; MEGF10, multiple EGF-like domains 10; OCR, oxygen consumption rate; OxPhos, oxidative phosphorylation; PI, pars intercerebralis; Pi3k, phosphoinositide 3-kinase.

resulting in insulin resistance, a state characterized by reduced cellular responsiveness to insulin [8]. Although HSD-induced insulin resistance has been well documented to occur in peripheral organs such as adipose tissue and the liver, less is known about whether HSD-induced insulin resistance also occurs in the brain. Despite the widespread expression of insulin receptors (IRs) in the brain, the brain has been historically regarded as insulin insensitive. This is largely because insulin is dispensable for glucose uptake in the brain [9] and its entry is limited by the blood–brain barrier [10,11]. However, growing evidence suggest that insulin exerts unique regulatory actions on the brain to control cognition, feeding, and systemic metabolism [12,13].

A key feature of neurodegenerative disorders is the diminished clearance of neuronal debris and neuron-secreted toxic proteins [14]. This can lead to inflammation, secondary neuronal death, and impaired axonal regeneration. Microglia are the brain's resident macrophages. When activated, they can swiftly mobilize to the site of disease or neuronal injury and initiate phagocytosis [15]. However, chronic activation of microglia can lead to the progressive decline of their phagocytic capacity as seen in the aging brain—the most at risk for neurodegenerative disorders [16]. Interestingly, obese humans and animals also display chronic activation of microglia, which has been shown to contribute to neuroinflammation [17]. However, little is known about the effects of obesity on glial phagocytic function. Uncovering whether and how diet-induced obesity disrupts glial phagocytosis may shed some light on the link between obesity and neurodegenerative disorders.

It was only recently discovered that microglia express IRs indicating that insulin can have a direct regulatory effect on microglial function [18,19]. However, the specific ways in which physiological factors, such as diet-induced alterations in insulin levels, modulate glial function remain unclear. Understanding these mechanisms can provide a deeper understanding of the impact of diet-induced obesity on glial cells and their potential contribution to neurodegenerative disorders.

Glial phagocytosis begins with the recognition of cellular debris via cell-surface receptors. Ablation of these receptors results in impaired clearance of cellular debris, while their overexpression leads to excessive neuronal pruning [20]. In *Drosophila*, a distinct subtype of glial cells known as ensheathing glia serves as the brain's resident phagocytes [21]. Analogous to mammalian microglia, ensheathing glia respond to neuronal injury by extending their membrane processes towards the site of damage and initiating phagocytic activity [22,23]. Just like their mammalian counterparts, *Drosophila* ensheathing glia, express phagocytic receptors, most prominently the mammalian multiple EGF-like domains 10 (MEGF10) homolog, Draper [24]. Also akin to microglia, ensheathing glia have shown proficiency in clearing transgenically expressed human amyloid beta (Aβ) via the phagocytic receptor Draper in a *Drosophila* model of Alzheimer's disease [25].

Several studies have demonstrated that baseline levels of Draper in the uninjured brain determine the phagocytic capacity of ensheathing glia [23,26]. Upon injury or disease, ensheathing glia up-regulate Draper [22,23,27,28]. However, low baseline levels may prevent Draper reaching a critical threshold for target detection leading to impaired clearance. Interestingly, Draper's baseline levels were found to be regulated by phosphoinositide 3-kinase (Pi3k), a downstream effector of IR signaling, while injury-induced Draper up-regulation is regulated by another insulin signaling downstream target, the transcription factor Stat92E [23]. Though local glial IR signaling has been shown to be a key regulator of Draper expression [29], it remains unknown whether obesogenic diets disrupt insulin signaling in glia and whether that disrupted signaling affects Draper expression and glial function.

Hence, we set out to address whether prolonged obesogenic diets in *Drosophila* disrupt glial phagocytic function. We have previously shown that prolonged HSD treatment causes

peripheral insulin resistance in adult flies [30]. Specifically, using a 30% HSD regime in adult flies that fed ad libitum, we were able to study the effect of chronic (>3 weeks) HSD exposure on feeding behavior. Here, using the previously established diet regime, we show that chronic HSD exposure leads to insulin resistance in ensheathing glia, which results in their impaired ability to clear axotomized olfactory neurons. Genetically inducing insulin release recapitulates HSD-induced Draper down-regulation, while attenuating insulin release rescues HSD-induced Draper down-regulation. Importantly, we show that genetically stimulating a downstream effector of IR signaling in ensheathing glia rescues HSD-induced insulin resistance and the down-regulation of Draper. Together, this study provides the first *in vivo* evidence of diet-induced regulation of glial phagocytic function.

## Results

### HSD affects the brain's metabolism and causes lipid droplet accumulation

We have previously established that adult *Drosophila* fed a prolonged HSD (see Methods) exhibit hallmarks of peripheral insulin resistance, including disrupted hunger responses [30]. Using the obesogenic 30% HSD diet paradigm (see Discussion) that we previously established and characterized [30], we sought to investigate the impact of prolonged HSD treatment on the brain's metabolic state. Cells rely on 2 main sources of energy: glycolysis, a series of cytosolic biochemical reactions to generate ATP, and mitochondrial oxidative phosphorylation (OxPhos). The interplay between OxPhos and glycolysis is tightly regulated to maintain a delicate balance [31,32] (Fig 1A). To assess the impact of HSD on the brain's OxPhos/glycolysis balance, we employed a functional *ex vivo* method to measure whole brain metabolism in adult fruit flies [33]. We measured the mitochondrial respiration rate (OCR) and the extracellular acidification rate (ECAR). OCR serves as a readout of OxPhos activity, while ECAR provides and indicator of glycolysis. We found that the brains of HSD-treated flies exhibited an elevation in OCR (Fig 1B) and a concomitant reduction in ECAR (Fig 1C), indicating a clear shift from glycolysis towards OxPhos (Fig 1D).

The olfactory system in flies has been established as a powerful model for glia–neuron interactions given its accessibility and well-defined histology. Therefore, we chose to focus on the antennal lobe region in this study to assess the effects of prolonged HSD treatment on glial function. Specifically, we focused on ensheathing glia, which are functionally similar to microglia and reside within the antennal lobe [22]. Given that microglial activation requires the metabolic switch from oxidative phosphorylation to glycolysis [34,35], we sought to examine the expression levels of the glycolytic enzyme lactate dehydrogenase (Ldh). Ldh is responsible for the final step of glycolysis, making it a reliable readout of glycolysis [36]. Notably, glycolysis is predominantly employed by glial cells within the brain [37–40]. Using a transgenic line with that expresses a fluorescent Ldh reporter [41,42], we began by subjecting these flies to either ND or HSD for 2 weeks, the time point at which the break in metabolic homeostasis occurs as revealed by our previous study [30]. Surprisingly, at 2 weeks of diet treatment, we did not observe a difference in Ldh levels between ND and HSD-fed flies (S1 Fig). However, at 3 weeks of diet treatment, the HSD-fed flies had significantly lower levels of Ldh in the antennal lobe region compared to the ND-fed flies (Fig 1E and 1F) suggesting attenuated glycolysis.

Human and animal studies have shown that microglia accumulate lipid droplets in aging and neurodegenerative disorders, leading to impaired function [43,44]. Given that glia are the main lipid storage units in the brain [44,45], and that their metabolic state affects their lipid storage [44], we asked whether HSD had an impact on glial lipid droplet formation. To answer this, we visualized the lipid droplets in the brains of ND and HSD-fed flies using LipidTOX, a neutral lipid stain, and drove membrane-tagged GFP expression specifically in ensheathing

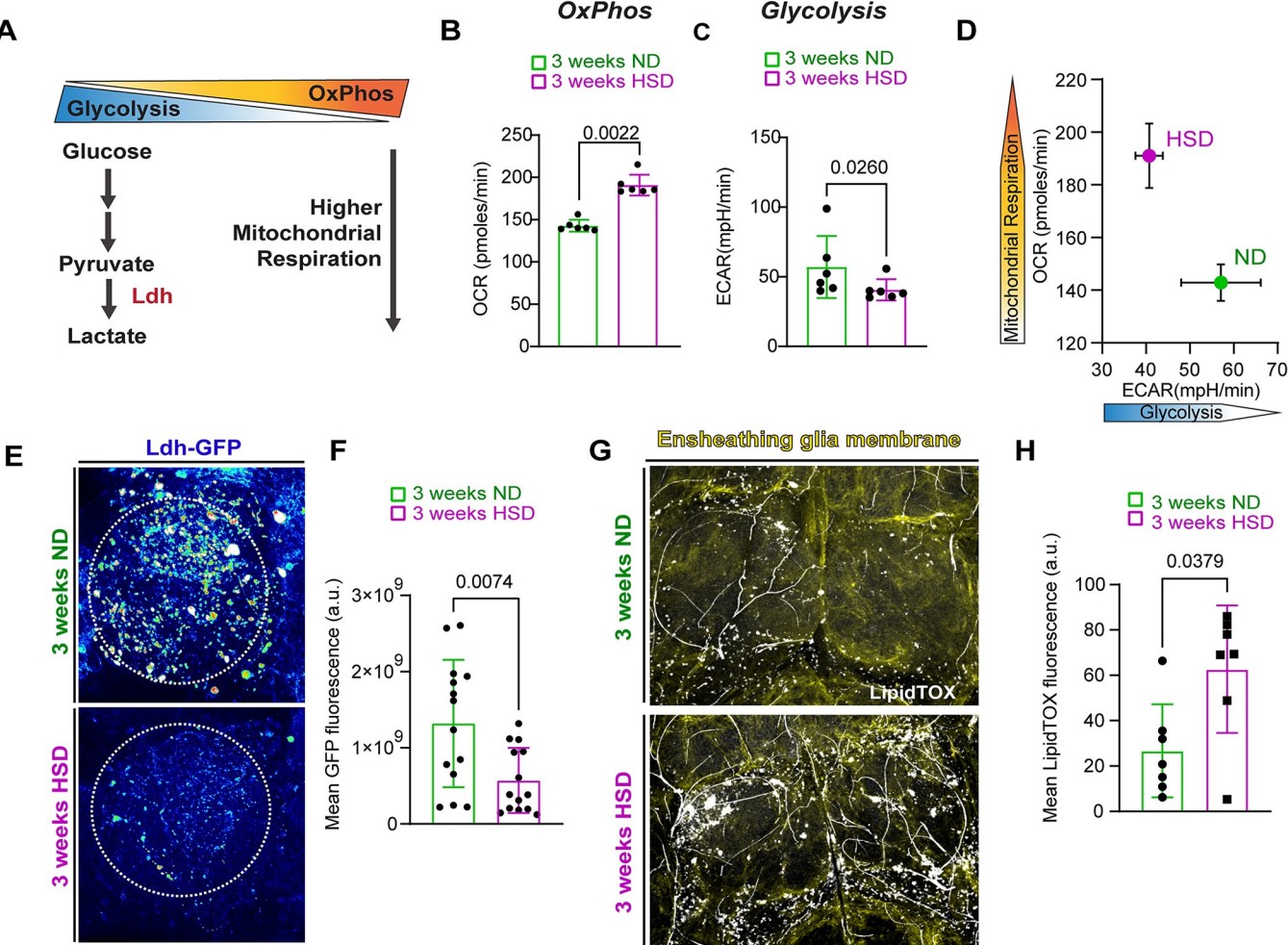

**Fig 1. HSD affects brain's metabolism and causes lipid droplet accumulation.** (A) Schematic showing the cellular energy balance between glycolysis and mitochondrial OxPhos. Attenuation of glycolysis results in the up-regulation of OxPhos. (B) Mean basal mitochondrial OCR collected from single adult brains of ND and HSD-fed flies using Agilent Seahorse XF HS Mini Analyzer. $N = 6$ measurement cycles collected from 3 brains/treatment group. Student's $T$ test with Welch's correction. (C) Mean basal ECAR collected from single adult brains of ND and HSD-fed flies using Agilent Seahorse XF HS Mini Analyzer. $N = 6$ measurement cycles collected from 3 brains/treatment group. Student's $T$ test with Welch's correction. (D) Mean ratio of brain OCR/ECAR of ND and HSD-fed flies showing that the brains of ND-fed flies exhibit more glycolysis, whereas the brains of HSD-fed flies lean more towards OxPhos. (E) Confocal images of Ldh-GFP in the antennal lobe region (dotted circle) of ND and HSD-fed flies. (F) Mean Ldh-GFP fluorescent intensity, obtained from Z-stack summation projections, within a defined region of interest (dotted circle) that covers the antennal lobe in ND and HSD-fed flies. Student's $T$ test with Welch's correction. $N$ = each circle represents an individual fly. (G) Confocal images of Lipidtox-stained (white) lipid droplets in the brains of ND and HSD-fed flies that express membrane-tagged GFP in their ensheathing glia (yellow). (H) Mean Lipidtox fluorescent intensity extracted from within the ensheathing glia mask. Student's $T$ test with Welch's correction. $N$ = each circle represents an individual fly. The data underlying this figure can be found in the supporting information file S1 Data. ECAR, extracellular acidification rate; HSD, high-sugar diet; OCR, oxygen consumption rate; OxPhos, oxidative phosphorylation.

glia. We found that prolonged HSD treatment markedly increased the fluorescent intensity of LipidTox within the ensheathing glia surrounding the antennal lobes (Fig 1G and 1H) indicating possible glial dysfunction.

## HSD causes insulin resistance in glia

Though lipid droplets can be neuroprotective in the appropriate context [46], lipid droplet accumulation and reduced glycolysis are tightly associated with impaired insulin signaling [25,43–46]. Lipid droplet accumulation and reduced glycolysis are tightly associated with

impaired insulin signaling [30,47–50]. While obesogenic diets have been established by us and others to cause peripheral insulin resistance [30,51,52], whether obesogenic diets lead to central brain insulin resistance remains unclear. In a previous study, our lab demonstrated that chronic HSD treatment of 2 weeks or more causes insulin resistance in the adult *Drosophila* adipose tissue [30]. To determine whether HSD treatment causes insulin resistance in the brain, we first examined the expression levels of the brain-specific gluconeogenic gene, glucose-6-phosphatase (G6P) [53]. G6P transcript levels are inversely correlated with insulin sensitivity [54–57]. Importantly, its elevated expression is associated with human and animal models of diabetes [58–61]. Through qPCR analysis of the heads of ND and HSD-fed flies, we found that HSD caused a 3.4-fold increase in G6P expression (Fig 2A). Furthermore, we looked at the expression levels of the IR substrate, Chico. Normally, when insulin signaling is low, the expression of the IR increases to enhance sensitivity [62]. We found that in response to HSD treatment, Chico was up-regulated by 2-fold (Fig 2B) supporting the idea that HSD causes brain insulin resistance.

Next, we compared the levels of the *Drosophila* insulin-like peptide 5 (Dilp5) retained in the insulin producing cells (IPCs) of ND and HSD-fed flies. Dilp5 is primarily produced by the IPCs and its secretion is dictated by nutrient abundance [63,64]. Dilp5's retention in the IPCs is often used as a readout of its secretion [65–70]. As expected, the HSD-fed flies showed reduced Dilp5 accumulation in their IPCs (Fig 2C and 2D), which was accompanied by increased Dilp5 transcript levels (Fig 2E) suggesting increased Dilp5 secretion.

To investigate whether the increase in insulin secretion led to glial insulin resistance, we used a transgenic line that expresses a fluorescent reporter (tGPH) for Pi3k [71,72], a downstream effector of IR signaling. Under normal conditions, the IR autophosphorylates upon interacting with insulin, which leads to activation of the Pi3k pathway. However, excessive levels of circulating insulin can attenuate IR sensitivity leading to reduced Pi3k activation [73,74] (Fig 2F). To this end, we subjected the tGPH flies to 3 weeks of either ND or HSD and measured tGPH fluorescence in the area surrounding the antennal lobe where ensheathing glia reside (Fig 2G and 2H). We found that HSD caused a significant down-regulation of Pi3k activity indicating insulin resistance (Fig 2G–2I). Notably, we did not observe any gross morphological defects in ensheathing glia in response to the prolonged HSD treatment (S2 Fig). To determine whether the HSD-induced attenuation of Pi3k signaling is specifically due to excessive systemic insulin secretion, we genetically induced insulin secretion from the IPCs by expressing the neuronal activator TrpA1 under the control of an IPC-specific Gal4 driver [66,70]. Remarkably, we found that forced activation of the IPCs for 1 week mimics the effects of a 3-week HSD exposure on Pi3k activity in ensheathing glia (Fig 2J–2L). In contrast, attenuating the release of insulin in the HSD-fed flies by expressing a genetically modified potassium channel (EKO) that inhibits neuronal activation [75,76] under the control of an IPC-specific driver increases glial insulin signaling (Fig 2M–2O). Together, these findings demonstrate that obesogenic diets directly cause glial insulin resistance through excess systemic insulin.

## HSD down-regulates basal Draper levels

As the brain's resident macrophages, microglia are crucial to the survival and function of the nervous system through their phagocytic activity [77]. Just like the mammalian microglia, the phagocytic activity of ensheathing glia is governed by the engulfment receptor, Draper (the *Drosophila* ortholog to the mammalian MEGF10) [22,28]. Given that Pi3k signaling has been shown to regulate basal Draper levels [23], we reasoned that HSD treatment and the ensuing down-regulation of Pi3k signaling (Fig 2G–2I) would result in reduced basal Draper levels. To test this, we measured Draper immunofluorescence within a subset of ensheathing glia in ND

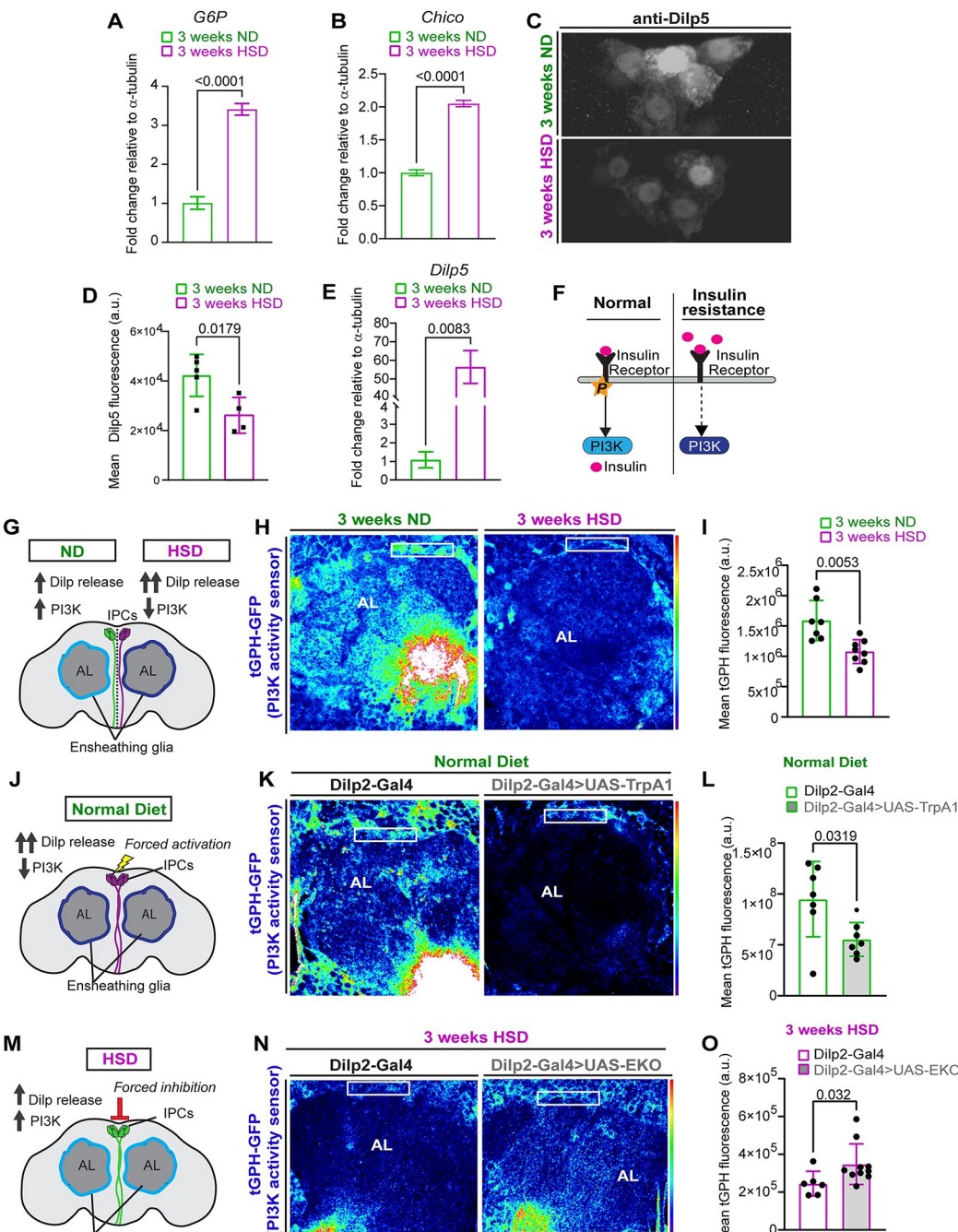

**Fig 2. HSD causes insulin resistance in glia.** (A) Mean fold change in *G6P* expression in ND and HSD-fed flies. *N* = 3 technical replicates of cDNA collected from 100 fly heads/treatment. Student *t* test with Welch's correction. (B) Mean fold change in *Chico* expression in ND and HSD-fed flies. *N* = 3 technical replicates of cDNA collected from 100 fly heads/treatment. Student *t* test with Welch's correction. (C) Representative z-stack summation projections of the IPCs immunostained with anti-Dilp5 of ND and HSD-fed flies. (D) Mean fluorescent intensity of anti-Dilp5 in the IPCs of ND and HSD-fed flies, obtained from a Z-stack summation projection that spans the entire depth of the IPCs. Student *t* test with Welch's correction. *N* = each circle represents an individual fly. (E) Mean fold change in *Dilp 5* expression in ND and HSD-fed flies. *N* = 3 technical replicates of cDNA collected from 100 fly heads/treatment. Student *t* test with Welch's correction. (F) Schematic of the insulin resistance model. Increased circulating insulin desensitizes the IRs leading to reduced Pi3k activity. (G) Schematic for experimental design and summarized results of (H and I). Flies fed an HSD exhibit reduced ensheathing glia Pi3k signaling despite increased Dilp release. (H) Representative confocal images of the antennal lobe region of flies expressing the Pi3k (tGPH) activity sensor that were fed either an ND or an HSD. (I) Mean fluorescent intensity of tGPH-GFP measured within a region of interest (white box) that coincides with the location of

ensheathing glia in ND and HSD-fed flies. Measurements were obtained from a Z-stack summation projection that spans the entire depth of the antennal lobe. Student *t* test with Welch's correction. *N* = each circle represents an individual fly. (J) Schematic for experimental design and summarized results of (K and L). Flies expressing TrpA1 specifically in their IPCs show reduced ensheathing glia Pi3k signaling similar to flies fed an HSD. (K) Confocal images of tGPH levels in the antennal lobe region of control flies and flies expressing TrpA1 in the IPCs that were fed an ND. TrpA1 expression force activates the IPCs to release Dilps. (L) Mean fluorescent intensity of GFP measured within a region of interest (white box) that coincides with the location of ensheathing glia in control and flies with Dilp2-driven TrpA1 expression. Measurements were obtained from a Z-stack summation projection that spans the entire depth of the antennal lobe. Student *t* test with Welch's correction. *N* = each circle represents an individual fly. (M) Schematic for experimental design and summarized results of (N and O). HSD-fed flies expressing EKO specifically in their IPCs show increased ensheathing glia Pi3k signaling compared to control flies. (N) Confocal images of tGPH levels in the antennal lobe region of control flies and flies expressing EKO in the IPCs that were fed an HSD. EKO expression attenuates the release of Dilps from the IPCs. (O) Mean fluorescent intensity of GFP measured within a region of interest (white box) that coincides with the location of ensheathing glia in control and flies with Dilp2-driven EKO expression. Measurements were obtained from a Z-stack summation projection that spans the entire depth of the antennal lobe. Student's *t* test with Welch's correction. *N* = each circle represents an individual fly. The data underlying this figure can be found in the supporting information file S2 Data. Dilp5, *Drosophila* insulin-like peptide 5; G6P, glucose-6-phosphatase; HSD, high-sugar diet; IPC, insulin-producing cell; IR, insulin receptor; Pi3k, phosphoinositide 3-kinase.

and HSD flies. At 2 weeks of diet treatment, we found no change in Draper levels (S3 Fig) consistent with unchanged Ldh levels (S1 Fig). However, we found that 3 weeks of HSD treatment caused a substantial reduction in basal Draper levels (Fig 3A–3C). Notably, another glial subtype in the cortex, which also expresses Draper, exhibits decreased basal Draper levels under HSD (S4 Fig). However, we opted to concentrate on ensheathing glia due to its well-established involvement in phagocytosis and the convenient experimental accessibility offered by the olfactory system. It is possible that HSD treatment results in non-insulin–dependent down-regulation of Draper signaling. Therefore, we reasoned that if glial insulin resistance (Fig 2G–2O) is responsible for Draper down-regulation in HSD-fed flies, then forced systemic insulin release from the IPCs would also result in Draper down-regulation. Indeed, we find that expressing the neuronal activator TrpA1 under the control of an IPC-specific Gal4 driver leads to reduced Draper levels under ND conditions (Fig 3D–3F). In contrast, attenuating systemic insulin release from the IPCs by expressing a genetically modified potassium channel (EKO) that inhibits neuronal activation paradoxically increases Draper expression in the HSD-fed flies (Fig 3G–3I).

Next, we reasoned that if glial insulin resistance underlies the down-regulation of Draper in HSD-fed flies, then stimulating Pi3k signaling, a downstream arm of insulin signaling, will up-regulate Draper expression. To address this, we expressed a constitutively active form of Pi3k that is fused to a farnesylation signal (CAAX) [78] under the control of an ensheathing glia promoter. We found that stimulating ensheathing glia Pi3k signaling increases Draper expression under HSD compared to the HSD-fed controls (Fig 3J–3L). Together, this suggests that HSD down-regulates Draper expression by inducing glial insulin resistance.

## HSD delays the clearance of degenerating axons by inhibiting injury-induced Draper and Stat up-regulation

Normally, neuronal injury triggers the up-regulation of Draper in ensheathing glia that peaks 1 day after injury and persists until neuronal debris has been cleared [28]. Therefore, we asked whether HSD treatment prevents the up-regulation of Draper after neuronal injury. To answer this, we took advantage of the accessibility of the olfactory neurons. We performed unilateral ablation of the third antennal segment, which houses the cell bodies of olfactory neurons. This results in the Wallerian degeneration of olfactory neurons' axons that project to the antennal lobe, which induces ensheathing glia to phagocytose axonal debris [22,28,79]. Then, we immunostained for Draper 1 day post antennal ablation (Fig 4A). As expected, Draper levels

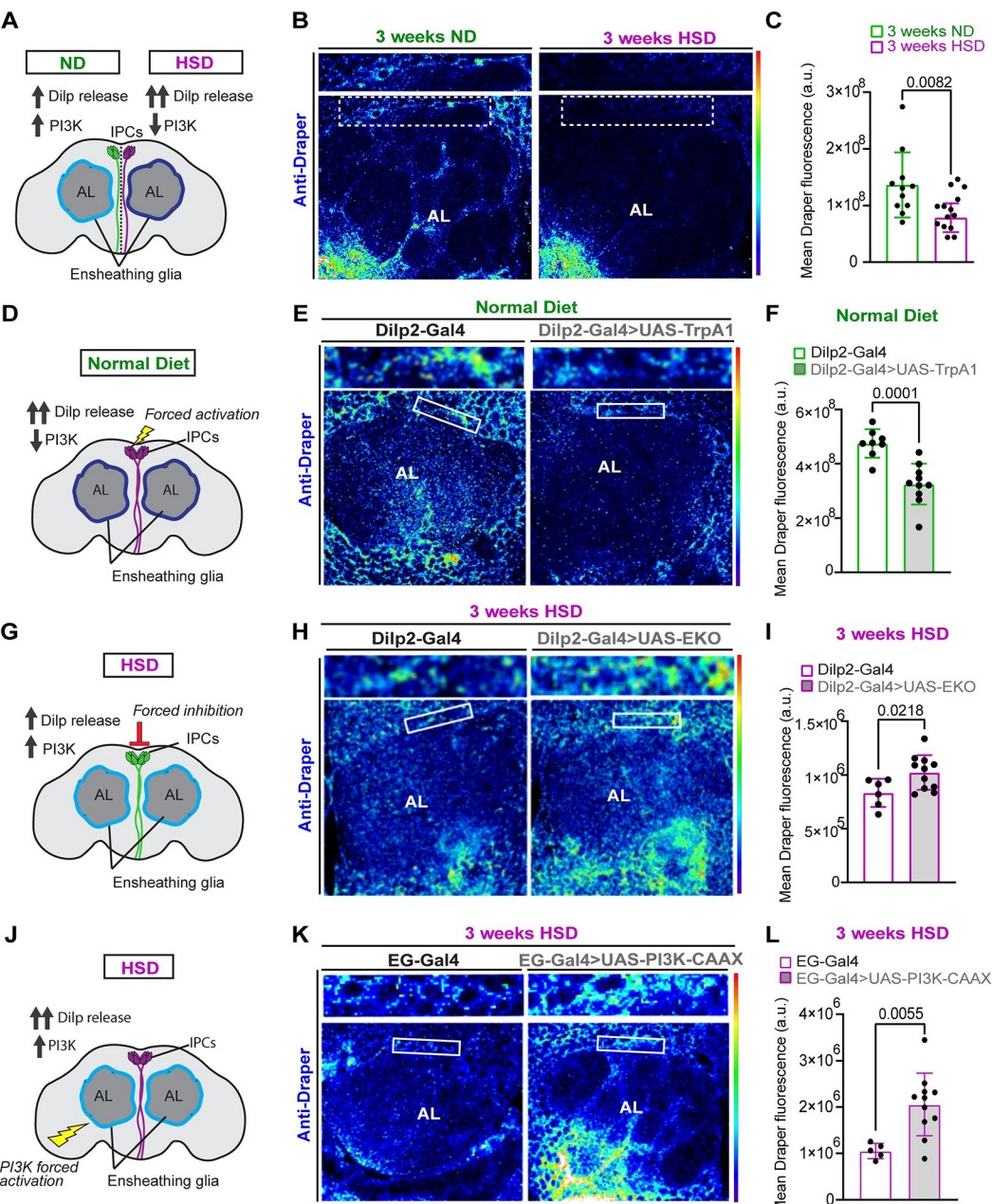

**Fig 3. HSD down-regulates basal Draper levels.** (A) Schematic for experimental design and summarized results of (B and C). Flies fed an HSD exhibit reduced Draper signaling despite increased Dilp release. (B) Confocal images of the antennal lobe region of flies fed an ND or an HSD that were immunostained with anti-Draper. (C) Mean fluorescent intensity of Draper measured within a region of interest (white box) that coincides with the location of ensheathing glia in ND and HSD-fed flies. Measurements were obtained from a Z-stack summation projection that spans the entire depth of the antennal lobe. Student *t* test with Welch's correction. *N* = each circle represents an individual fly. (D) Schematic for experimental design and summarized results of (E and F). Flies expressing TrpA1 specifically in their IPCs show reduced Draper signaling similar to flies fed an HSD. (E) Confocal images of Draper levels in the antennal lobe region of control flies and flies expressing TrpA1 in the IPCs that were fed an ND. TrpA1 expression force activates the IPCs to release Dilps. (F) Mean fluorescent intensity of Draper measured within a region of interest (white box) that coincides with the location of ensheathing glia in control and flies with Dilp2-driven TrpA1 expression. Measurements were obtained from a Z-stack summation projection that spans the entire depth of the antennal lobe. Student *t* test with Welch's correction. *N* = each circle represents an individual fly. (G) Schematic for experimental design and summarized results of (H and I). HSD-fed flies expressing EKO specifically in their IPCs show increased Draper signaling compared to control flies. (H) Confocal images of Draper levels in the antennal lobe region of control flies and flies expressing EKO in the IPCs that were fed an HSD. EKO expression attenuates the release of Dilps from the IPCs. (I) Mean fluorescent intensity of Draper measured

within a region of interest (white box) that coincides with the location of ensheathing glia in control and flies with Dilp2-driven EKO expression. Measurements were obtained from a Z-stack summation projection that spans the entire depth of the antennal lobe. Student *t* test with Welch's correction. *N* = each circle represents an individual fly. (J) Schematic for experimental design and summarized results of (K and L). HSD-fed flies expressing a constitutively active form of Pi3k specifically in their IPCs show increased ensheathing glia Draper signaling compared to control flies. (K) Confocal images of Draper levels in the antennal lobe region of HSD-fed control flies and flies expressing Pi3k-CAAX in ensheathing glia. (L) Mean fluorescent intensity of Draper measured within a region of interest (white box) that coincides with the location of ensheathing glia. Measurements were obtained from a Z-stack summation projection that spans the entire depth of the antennal lobe. Student's *T* test with Welch's correction. *N* = each circle represents an individual fly. The data underlying this figure can be found in the supporting information file S3 Data. HSD, high-sugar diet; IPC, insulin-producing cell; Pi3k, phosphoinositide 3-kinase.

increased significantly in the ND-fed flies, whereas the HSD-fed flies showed no up-regulation in Draper (Fig 4B and 4C).

While baseline levels of Draper are regulated by Pi3k [23], it has been shown that Stat92E, a transcription factor that acts downstream of both Draper and insulin signaling is essential for the injury-induced Draper up-regulation [23,29]. Draper-dependent activation of Stat92E creates a positive autoregulatory loop in which Stat92E up-regulates the transcription of the *Draper* gene [23]. Given that HSD treatment causes a diminished injury-induced Draper response (Fig 4B and 4C), we reasoned that Stat92E signaling would be attenuated in the antennal lobe region of the HSD-fed flies. To address this, we used a transgenic reporter line with 10 Stat92E binding sites that drive the expression of a destabilized GFP [23]. We found that while ND-fed flies exhibited the expected post injury Stat92E up-regulation, HSD treatment caused a reduction in Stat92E levels at baseline and inhibited post injury up-regulation (Fig 4D and 4E). Together, these data indicate that chronic HSD attenuate the Stat92E/Draper signaling pathway leading to impaired glial phagocytotic function.

To understand how disrupted Draper signaling in the HSD-fed flies affect glial phagocytic function, we subjected flies that express membrane tagged GFP in a subset of olfactory neurons (Odorant receptor 22a) to either 3 weeks of ND or HSD, then performed unilateral antennal ablation and examined the rate of GFP clearance over time (Fig 4F). By normalizing GFP fluorescence on the injured side to the uninjured side of the same animal, we were able to establish an endogenous control (Fig 4G). We found that HSD-fed flies had higher levels of GFP florescence at every time point indicating a delay in axonal clearance (Fig 4G and 4H). Together, this data indicate that diet-induced glial insulin resistance impairs the clearance of neuronal debris by down-regulating Draper.

## Discussion

With increased life expectancy, age-related neurodegenerative disorders are expected to rise, placing a tremendous burden on the healthcare system [5–7]. Large-scale epidemiological studies have found that mid-life obesity is an independent risk factor for developing neurodegenerative disorders [1–3]. However, the mechanism underlying this connection remains unclear. Here, using a *Drosophila* in vivo model, we draw a causal link between diet-induced obesity and impaired glial phagocytic function, a major contributor to the pathology of age-related neurodegenerative disorders [80]. We show that excessive systemic insulin signaling leads to glial insulin resistance, which dampens the expression of the engulfment receptor, Draper, resulting in impaired glial clearance of degenerating axons. Together, our study provides a strong mechanistic insight into how diet-induced obesity alters glial function, thereby increasing the risk of neurodegenerative disorders.

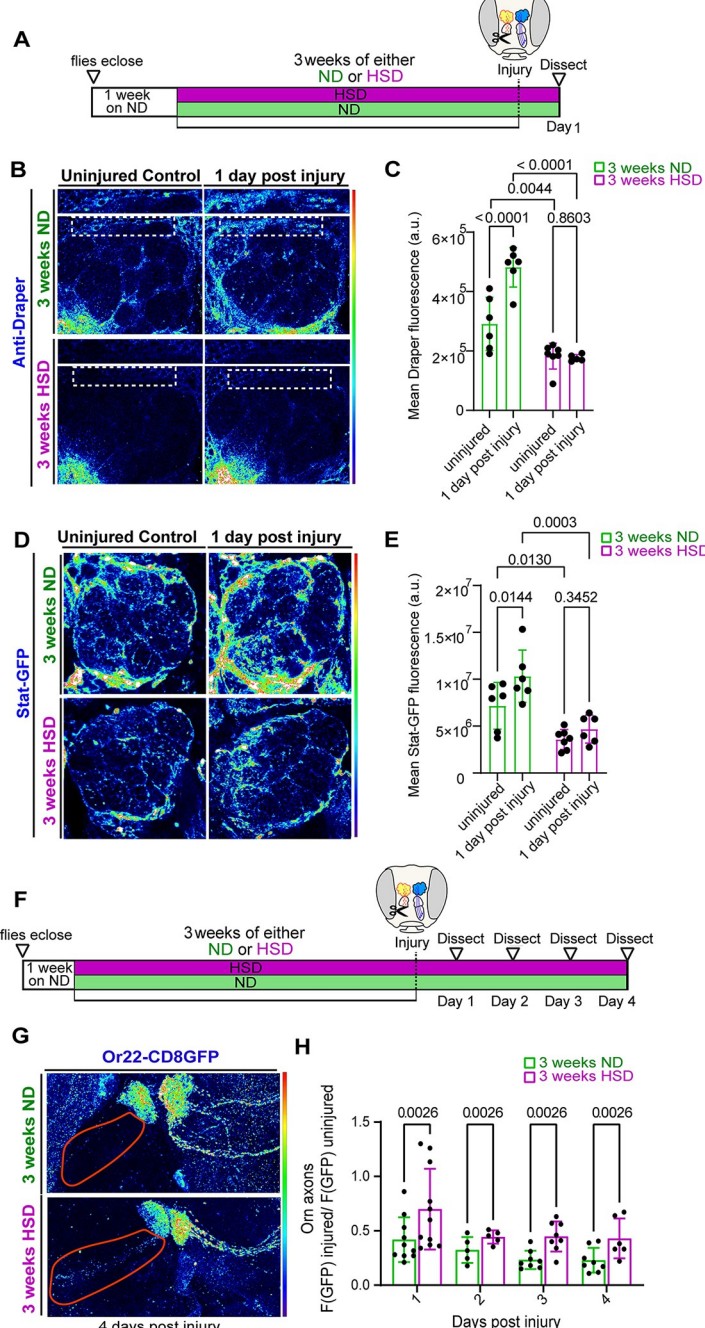

**Fig 4. HSD inhibits injury-induced Stat up-regulation and delays clearance of degenerating axons.** (A) Schematic showing the experimental strategy for (B–E). Flies were fed an ND for 1 week post eclosion before being flipped into either ND or HSD for 3 weeks. Unilateral antennal ablation was performed and Draper and Stat levels were assayed 24 h later. (B) Confocal images of Draper levels in the antennal lobe region of uninjured and injured ND or HSD-fed flies. (C) Mean fluorescent intensity of Draper measured within a region of interest (white box) that coincides with the location of ensheathing glia in ND and HSD-fed flies. Measurements were obtained from a Z-stack summation projection that spans the entire depth of the antennal lobe. Student *t* test with Welch's correction. *N* = each circle represents an individual fly. (D) Confocal images of Stat92E levels in the antennal lobe region of uninjured and injured ND or HSD-fed flies. (E) Mean fluorescent intensity of GFP measured within a region of interest (white box) that covers the antennal lobe in ND and HSD-fed flies. Measurements were obtained from a Z-stack summation projection that spans the entire depth of the antennal lobe. Two-way ANOVA. *N* = each circle represents an individual fly. (F) Schematic showing the experimental strategy for (B and C). Flies expressing GFP in a subset of their olfactory neurons were fed an ND for 1 week post eclosion then flipped into either ND or HSD for 3 weeks. Unilateral antennal ablation

was performed on day 21 of diet treatment and GFP levels were assayed on days 1–4 post injury. (G) Confocal images of the antennal lobe region of flies expressing membrane-tagged GFP in a subset of olfactory neurons. See S1 and S2 Videos. (H) Mean fluorescent intensity of GFP obtained from z-stack summation projections. Measurements were taken overtime within a region of interest (axons; red outline) that coincides with the location of olfactory axons in ND and HSD-fed flies. The data underlying this figure can be found in the Supporting information file S4 Data. HSD, high-sugar diet.

## HSD causes glial insulin resistance

Insulin signaling is critical for cell metabolism and function. However, excess systemic insulin can lead to insulin resistance, which results in diminished cellular response. Obesogenic diets are well known to cause insulin resistance in peripheral tissues, including fat, which rely on insulin to regulate glucose uptake [73,81]. However, the evidence for diet-induced insulin resistance in the brain is scarce. This is mainly due to the belief that the brain is insulin independent [9], though evidence suggests that it acts on neurons and glia in a glucose-independent manner [82].

In our studies, we have adopted a 30% HSD regime. The rationale for this stems from prior studies showing that larvae reared on HSDs develop insulin resistance and diabetic phenotypes. Musselman and colleagues found that isocaloric diets high in protein and fat did not affect insulin signaling [51]. According to this study, insulin resistance phenotypes were associated with an increase in sugar intake but not with an increase in calorie intake [51]. Consistent with this, we observed peripheral insulin resistance in adult *Drosophila* raised on a 30% HSD [30]. It is worth noting, however, that other obesogenic regimes in *Drosophila*, such as coconut oil-based high-fat diets (HFDs), have been used to explore time-restricted feeding's effect on obesogenic decline [5,83]. Given that different obesogenic regimes are likely to have varied impacts, work will be needed to assess whether an HFD regime causes glial insulin resistance in *Drosophila*.

Microglia play a significant role in maintaining nervous system homeostasis and their dysfunction is implicated in a myriad of metabolic and neurodegenerative disorders [77,84,85]. Although microglia express the IR [18], it remains unknown whether obesogenic diets can result in glial insulin resistance. In this study, we showed that prolonged HSD treatment attenuated glial Pi3k signaling, a downstream arm of the IR signaling pathway. This coincided with reduced ECAR and Ldh levels indicating depressed glycolysis, which is tightly regulated by insulin signaling [47,49,50]. Consistent with this, insulin resistance is associated with reduced glycolysis [86,87]. Interestingly, a prerequisite to microglial activation is the metabolic switch from oxidative phosphorylation to glycolysis [34,35]. This suggests that HSD-induced insulin resistance may prevent glial activation by disrupting their OxPhos-glycolysis balance. Because Ldh expression is an indirect readout of glycolysis, more precise metabolic analysis is needed to determine the specific glial metabolic defects caused by HSD. Furthermore, it is important to acknowledge that both the Pi3k and Ldh reporters are non-cell specific. Therefore, it is likely that HSD is causing a down-regulation of Pi3k signaling and glycolysis in both glia and neurons. Given that glia and neurons are intricately connected, it is possible that dysfunctional neuronal metabolism further exacerbates glial dysfunction.

There are 7 *Drosophila* insulin-like peptides (Dilps). Dilps 2, 3, and 5 are primarily produced by the IPCs, which reside in the pars intercerebralis (PI) region of the *Drosophila* brain —the invertebrate equivalent to the mammalian hypothalamus. In the adult fly brain, the IPCs terminate their axons on peripheral targets including the gut and aorta for systemic Dilp release [64,88,89]. It is possible that excess circulating Dilps induced by HSD cause glial insulin resistance indirectly by dysregulating peripheral organs, such as muscles, guts, and adipose tissue. Nevertheless, distinct IPC arborizations have been observed in the brain, specifically in

the tritocerebrum proximal to the antennal lobes, raising the possibility that the IPCs act in a paracrine manner. Given that genetic activation and inhibition of the IPCs for a short period of time was sufficient to influence glial Pi3k signaling, it is possible that IPC-released Dilps act directly on ensheathing glia. In the future, it would be interesting to untangle the IPC-ensheathing glia insulin signaling circuit.

### HSD-induced glial dysfunction resembles that caused by aging

A hallmark of neurodegenerative disorders is the failure to clear neuronal debris and cytotoxic proteins, triggering a cascade of devastating effects that include inflammation, cell death, and impaired regeneration. Therefore, it is not surprising that microglial dysfunction is implicated in driving the pathogenesis of many neurodegenerative disorders [15]. Although it is known that microglia express the IRs and respond to insulin treatment in vitro [19,90], it remained unknown whether they experience insulin resistance and whether that impacts their phagocytic activity. Both obesity and age-related neurodegenerative disorders are associated with dysfunctional insulin signaling [73,81,91,92], suggesting a potential link. Here, we show that diet-induced insulin resistance disrupts glial phagocytic activity by down-regulating the phagocytic receptor, Draper. We were able to demonstrate the direct effects of insulin signaling by showing that modulating Dilp release alone mimicked HSD-induced glial defects. While other groups have shown that local glial IR activity regulates Draper expression [29], we found that systemic insulin signaling directly regulates glial Draper expression.

It has been demonstrated by other groups as well as us that physiological factors affect glial function in *Drosophila*. Stanhope and colleagues [93] found that sleep plays a crucial role in glial phagocytosis. Similar to HSD treatment, sleep loss causes Draper to down-regulate, which leads to a failure to clear neuronal debris after injury. As with obesity and type 2 diabetes, dysregulated sleep has been associated with neurodegenerative disorders, further supporting the link between dysfunctional glial phagocytosis and neurodegenerative disorders. It is interesting to draw parallels between the results of this study and a recent study by Purice and colleagues [26] that found that aged flies exhibit delayed axonal clearance because of impaired Draper and Pi3k activity. The connection between HSD exposure and aging holds true for humans and mammals as well. Age and neurodegenerative disorders are associated with reductions in MEGF10 [94], the mammalian ortholog of Draper, impaired glial phagocytosis [16], and stunted glycolysis [95]. One study showed that insulin infusion into young rats activated microglia, but this effect was not observed in older rats suggesting that microglia's insulin sensitivity is age dependent [96]. Furthermore, similar to our findings, which show an accumulation of lipids in the brains of HSD-fed flies, aged microglia also accumulate lipid droplets leading to impaired phagocytic function [44]. As a result, it could be argued that chronic HSD exposure may "accelerate" aging in flies. It would be interesting to explore this connection in more detail by comparing the transcriptomes of aged and HSD-treated flies in the future.

## Methods

### Fly husbandry

The following *Drosophila* strains were used: Or22a-mCD8GFP (BDSC #52620), Ensheathing glia-Gal4 (BDSC # 39157), Cortex glia-Gal4 (BDSC#45784), 10xStat92E-GFP (Perrimon Lab), Ldh-GFP (a generous gift from Dr. Tennessen), tGPH (Gift of Bruce Edgar), Dilp2-Gal4 (a gift from P. Shen), UAS-TrpA1 (BDSC #26263), UAS-EKO22 (BDSC #40974), UAS-Pi3k-CAAX (BDSC #8294). Flies were housed in 25°C incubators and all experiments were done on adult male flies. To induce the expression of the temperature sensitive TrpA1, flies were moved to a 29°C incubator 1 week post eclosion. Following 1 week after eclosion, the flies were placed on either a normal diet

containing 15 g yeast, 8.6 g soy flour, 63 g corn flour, 5 g agar, 5 g malt, 74 mL corn syrup per liter, or an HSD, which consists of an additional 300 g of sucrose per liter (30% increase).

## Whole brain oxygen consumption rate (OCR) and extracellular acidification rate (ECAR)

OCR and ECAR were obtained using Agilent Seahorse XF HS Mini Analyzer as previously described [33]. Prior to the experiment, an Agilent Seahorse cartridge was hydrated overnight at 25˚C with 200 µl of calibrant solution from Agilent. The following day, brains from adult flies were dissected in phosphate buffered solution (PBS) then immediately transferred into an 8-well cell plate from Agilent. Wells 1 and 6 were used as negative controls containing only 200 µl of Agilent Seahorse assay media supplemented with 10 mM glucose and 10 mM sodium pyruvate. Using dull forceps, each brain was positioned at the bottom of the well, centrally located between 3 raised spheres. Three brains per condition were used as it is the maximum number of samples that can be loaded within the same plate. The tissue restraints were gently lowered down using dull forceps. The tissue restraints were designed by Neville and colleagues [33] and manufactured by the Instrument Design and Fabrication Core Facility at Arizona State University. Basal OCR and ECAR measurements were collected from 6 cycles.

## Antennal nerve injury

As adapted from [22,23,26–29,79,93,97], flies were anesthetized using CO2 and antennal nerve injury was accomplished by unilaterally removing the third antennal segment of anesthetized adult flies using forceps. Flies were then placed back into either ND or HSD until they were dissected 24 h after injury or as indicated otherwise in the figure legends.

## Immunostaining

Immunostaining of adult brains and fat bodies was performed as previously described [65,66]. Tissues were dissected in ice-cold PBS. Brains were fixed overnight in 0.8% paraformaldehyde (PFA) in PBS at 4˚C. The fixed brains were washed 5 times in PBS with 0.5% BSA and 0.5% Triton X-100 (PAT), blocked for 1 h in PAT + 5% NDS, and then incubated overnight at 4˚C with the primary antibodies. Following incubation, the brains were washed 5 times in PAT, re-blocked for 30 min, and then incubated in secondary antibody in block for 4 h at room temperature. Finally, the brains were washed 5 times in PAT, then mounted on slides in Slow fade gold antifade. Primary antibodies were as follows: rabbit anti-Dilp5 (1:500; this study), Chicken anti-GFP (1:500; Cat# ab13970, RRID:AB_300798), and Mouse anti-Draper (1:50; DSHB 5D14 RRID:AB_2618105). Secondary antibodies from Jackson ImmunoResearch (1:500) include donkey anti-Chicken Alexa 488 (Cat# 703-545-155, RRID: AB_2340375) and donkey anti-mouse Alexa 594 (Cat# 715-585-150, RRID:AB_2340854). Lipid droplets were stained with LipidTox (1:500, Thermo Fisher Cat#H34477) overnight at 4˚C.

## Image analysis

Images were acquired with a Zeiss LSM 800 confocal system and analyzed using ImageJ [98]. All images within each experiment were acquired with the same confocal settings. Z-stack summation projections that spanned the depth of the antennal lobes at 0.3 µm intervals were generated and a region of interest (indicated on the fig) was used to measure the fluorescent intensity of GFP or Draper. Ldh fluorescent intensity was measured within an ROI that covered the entirety of the left antennal lobe. Draper, Stat92, and Pi3k fluorescent intensity were performed within an ROI in the dorsal medial antennal lobe membrane as previously

described [26,28,93]. Cortex glia's Draper fluorescent intensity was measured by drawing a region of interest dorsal to the antennal lobe (see representative ROI in S2 Fig). The size and dimensions of all ROIs were maintained consistently throughout each experiment. Dilp 5 was measured using z-stack summation projections that included the full depth of the IPCs. A region of interest around the IPCs was manually drawn using the free hand tool and the integrated density values were acquired.

## Lipid content analysis

The image analysis pipeline to quantify lipid content in the ensheathing glia surrounding antennal lobes was developed in MATLAB R2022a. To exclude the tracheal structures from the lipid quantification, a morphological filter was applied to the image set: the LipidTox signal was first binarized in 3D using the Otsu method, and the length of the 3 principal axes as well as the volume of each resulting binary object were collected. To separate small, round-like structures (lipid droplets) from long, tubular structures (trachea), the ratio between the longest and the shortest principal axis lengths was measured for each object, and objects with a length ratio greater than 5 or with a volume greater than 5,000 voxels were filtered out. Visual inspections confirmed that these filtering parameters removed most, if not all of the tracheal structures while keeping most of the lipid droplet objects. The boundaries of the antennal lobes were manually drawn from the 2D maximal projection of the GFP channel labeling the ensheathing glia, and these boundaries were propagated along the z-axis to create a 3D volume containing the antennal lobes. The integral LipidTox pixel intensities of each lipid object were then extracted from within the ensheathing glia mask.

## qPCR

One hundred fly heads were dissected in RNAlater, and then placed in 30 μl of TriReagent and a scoop of beads in a 1.5 mL safelock tube. The heads were homogenized using a bullet blender. RNA was then isolated using a Direct-zol RNA microprep kit following the manufacturer's instructions. Isolated RNA was synthesized into cDNA using the Bio-Rad iScript RT supermix for RT-qPCR and qPCR was performed using the Bio-Rad ssoAdvanced SYBR green master mix. Primers used are as follows: *alpha-tubulin* (endogenous control), forward: AGCGGTAGTGTCT GCCGTGT and reverse: CCAGCGTGGATTTGACCGGA; *Dilp5*, forward: GCTCCGAATCTC ACCACATGAA and reverse: GGAAAAGGAACACGATTTGCG; *G6P*, forward: TGAGTTTTG GAAGGCCCCTTT and reverse: CACGAGGATTTCCTGACTGAAG; *Chico*, forward: AAGAA GTTCCTGCAAAGAGCC and reverse: CCAAACGGCGATTGATGTTGA. Relative quantification of mRNA was performed using the comparative CT method and normalized to *alpha-tublin* mRNA expression. Three technical replicates used for each gene.

## Statistics

All the statistical tests were done using GraphPad PRISM (GraphPad Software Incorporated, La Jolla, Ca, USA, RRID:SCR_002798). The assumption of normality was tested using Shapiro–Wilk's test. A *t* test was performed for Figs 1–3, and S1, S3, and S4 Figs, while a 2-way ANOVA was used for Fig 4. Significance was set at $p < 0.05$. Data are presented as means ± standard deviation. *N* for each experiment is detailed in the figs.

## Supporting information

**S1 Fig.** (A) Confocal images of Ldh-GFP in the antennal lobe region (dotted circle) of ND and HSD-fed flies after 2 weeks of diet treatment. (B) Mean Ldh-GFP fluorescent intensity,

obtained from Z-stack summation projections, within a defined region of interest (dotted circle) that covers the antennal lobe in ND and HSD-fed flies. The data underlying this figure can be found in the Supporting information file S5 Data.
(TIF)

**S2 Fig. The antennal lobe regions of flies with ensheathing glia-driven membrane tagged GFP.** No observable gross morphological defects in the HSD-fed flies.
(TIF)

**S3 Fig.** (A) Confocal images of the antennal lobe region of flies fed an ND or an HSD for 2 weeks that were immunostained with anti-Draper. (B) Mean fluorescent intensity of Draper measured within a region of interest (white box) that coincides with the location of ensheathing glia in ND and HSD-fed flies. Measurements were obtained from a Z-stack summation projection that spans the entire depth of the antennal lobe. Student *t* test with Welch's correction. *N* = each circle represents an individual fly. The data underlying this figure can be found in the Supporting information file S6 Data.
(TIF)

**S4 Fig.** (A) Representative confocal images of the antennal lobe region of flies expressing membrane GFP under the control of a cortex glia-specific Gal4 driver. Draper immunostaining colocalizes with cortex glia. The yellow box represents a sample ROI in which Draper fluorescence was measured. (B) Confocal images of the antennal lobe region of flies fed an ND or an HSD for 2 weeks that were immunostained with anti-Draper. (C) Mean fluorescent intensity of Draper measured within a region of interest (white box) that coincides with the location of cortex glia in ND and HSD-fed flies. Measurements were obtained from a Z-stack summation projection that spans the entire depth of the antennal lobe. Student *t* test with Welch's correction. *N* = each circle represents an individual fly. The data underlying this figure can be found in the Supporting information file S7 Data.
(TIF)

**S1 Video.** 3D rendering of uninjured (right) and axotomized (left) olfactory neurons in the ND-fed flies. Note complete clearance of axonal debris by day 4.
(AVI)

**S2 Video.** 3D rendering of uninjured (right) and axotomized (left) olfactory neurons in the HSD-fed flies. Note incomplete clearance of axonal debris by day 4.
(AVI)

**S1 Data. Sheet 1. OCR measurements of the brains of ND and HSD-fed flies, referenced in Fig 1B.** Sheet 2. ECAR measurements of the brains of ND and HSD-fed flies, referenced in Fig 1C. Sheet 3. Ratio of OCR to ECAR, referenced in Fig 1D. Sheet 4. Mean Ldh-GFP fluorescent intensity in the antennal lobes region of ND and HSD-fed flies, referenced in Fig 1F. Sheet 5. Mean LipidTox fluorescent intensity in the ensheathing glia of ND and HSD-fed flies, referenced in Fig 1H.
(XLSX)

**S2 Data. Sheet 1. Fold change in G6P transcript levels in the brains of ND and HSD-fed flies, referenced in Fig 2A.** Sheet 2. Fold change in Chico transcript in the brains of ND and HSD-fed flies levels, referenced in Fig 2B. Sheet 3. Mean Dilp5 fluorescent intensity in the brains of ND and HSD-fed flies, referenced in Fig 2D. Sheet 4. Fold change in Dilp5 transcript levels in the brains of ND and HSD-fed flies, referenced in Fig 2E. Sheet 5. Mean tGPH fluorescent intensity in the ensheathing glia of ND and HSD-fed flies, referenced in Fig 2I. Sheet 6.

Mean tGPH fluorescent intensity in the ensheathing glia of ND-fed wild type and flies expressing TrpA1 specifically in their IPCs, referenced in Fig 2L. Sheet 7. Mean tGPH fluorescent intensity in the ensheathing glia of HSD-fed wild type and flies expressing EKO specifically in their IPCs, referenced in Fig 2O.
(XLSX)

**S3 Data. Sheet 1. Mean Draper fluorescent intensity in the ensheathing glia of ND and HSD-fed flies, referenced in Fig 3C. Sheet 2.** Mean Draper fluorescent intensity in the ensheathing glia of ND-fed wild type and flies expressing TrpA1 specifically in their IPCs, referenced in Fig 3F. Sheet 3. Mean Draper fluorescent intensity in the ensheathing glia of HSD-fed wild type and flies expressing EKO specifically in their IPCs, referenced in Fig 3I. Sheet 4. Mean Draper fluorescent intensity in the ensheathing glia of HSD-fed wild type and flies expressing a constitutively active form of Pi3k specifically in their IPCs, referenced in Fig 3L.
(XLSX)

**S4 Data. Sheet 1. Mean Draper fluorescent intensity in the ensheathing glia of ND and HSD-fed flies with 1 day after unilateral antennal ablation, referenced in Fig 4C.** Sheet 2. Mean Stat-GFP fluorescent intensity in the ensheathing glia of ND and HSD-fed flies with 1 day after unilateral antennal ablation, referenced in Fig 4E. Sheet 3. Mean membrane GFP fluorescent intensity, referenced in Fig 4H.
(XLSX)

**S5 Data. Sheet 1. Mean Ldh-GFP fluorescent intensity in the antennal lobes region after 2 weeks of either ND or HSD treatment, referenced in S1B Fig.**
(XLSX)

**S6 Data. Sheet 1. Mean Draper fluorescent intensity in the ensheathing glia after 2 weeks of either ND or HSD treatment, referenced in S3B Fig.**
(XLSX)

**S7 Data. Sheet 1. Mean Draper fluorescent intensity in cortex glia after 3 weeks of either ND or HSD treatment, referenced in S4C Fig.**
(XLSX)

## Acknowledgments

We thank Dr. Jason M Tennessen for generously donating the Ldh-GFP transgenic fly line used in this article. We also thank Dr. Marla Tipping for sharing her tissue restraints designs and her valuable input on the ex vivo whole brain metabolic measurements Seahorse. We thank Dr. Lucas Sullivan, in the Human Biology Division at the Fred Hutch, for training and access to the Agilent Seahorse HF Flux analyzer. Dr. Julien Dubrulle in the Cellular Imaging Shared Resources at the Fred Hutch for support on the Lipid content analysis.

## Author Contributions

**Conceptualization:** Mroj Alassaf, Akhila Rajan.

**Data curation:** Mroj Alassaf.

**Formal analysis:** Mroj Alassaf.

**Funding acquisition:** Akhila Rajan.

**Investigation:** Mroj Alassaf, Akhila Rajan.

**Project administration:** Akhila Rajan.

**Supervision:** Akhila Rajan.

**Writing – original draft:** Mroj Alassaf.

**Writing – review & editing:** Akhila Rajan.

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
