## [Editor Report · Decision Letter 0]

24 Mar 2023

Dear Dr Rajan, 

Thank you for submitting your manuscript entitled "Diet-Induced Glial Insulin Resistance Impairs The Clearance Of Neuronal Debris" for consideration as a Short Report by PLOS Biology. I apologize for our delay in sending you an initial decision - I have been traveling and at a conference for the last week, and so am a bit slower than normal. 

Your manuscript has now been evaluated by the PLOS Biology editorial staff as well as by an academic editor with relevant expertise and I am writing to let you know that we would like to send your submission out for external peer review.

Once your full submission is complete, your paper will undergo a series of checks in preparation for peer review. After your manuscript has passed the checks it will be sent out for review. To provide the metadata for your submission, please Login to Editorial Manager (https://www.editorialmanager.com/pbiology) within two working days, i.e. by Mar 28 2023 11:59PM.

Kind regards,

Luke

Lucas Smith, Ph.D.

Associate Editor

PLOS Biology

lsmith@plos.org

---

## [Decision Letter · Decision Letter 1]

27 Apr 2023

Dear Dr Rajan,

Thank you for your patience while your manuscript "Diet-Induced Glial Insulin Resistance Impairs the Clearance Of Neuronal Debris" was peer-reviewed at PLOS Biology. It has now been evaluated by the PLOS Biology editors, an Academic Editor with relevant expertise, and by several independent reviewers. In light of the reviews, which you will find at the end of this email, we would like to invite you to revise the work to thoroughly address the reviewers' reports.

Given the extent of revision needed, we cannot make a decision about publication until we have seen the revised manuscript and your response to the reviewers' comments. Your revised manuscript is likely to be sent for further evaluation by all or a subset of the reviewers.

**IMPORTANT - SUBMITTING YOUR REVISION**

*Re-submission Checklist*

*Published Peer Review*

*PLOS Data Policy*

*Blot and Gel Data Policy*

Sincerely,

Lucas

Lucas Smith, Ph.D.

Associate Editor

PLOS Biology

lsmith@plos.org

REVIEWS:

Reviewer #1: This manuscript examines the interactions between diet and glial debris clearance, using a well-established model of Wallerian degeneration in Drosophila. Previously research groups have found that this process is regulated by insulin, aging, and sleep. It is therefore very surprising that the connection to diet has not been investigated. This line of investigation is potentially highly impactful because of links between dietary dysregulation of insulin and negative public health outcomes. In addition, the manuscript provides a direct link between high-sugar diet and changes within glial cells. Therefore, it is of broad interest to researchers studying glial biology, diet and metabolism, and aging. The experiments are well presented and elegantly straight-forward. The statistical analyses are appropriate and the experiments are well-controlled. The data is presented in a way that is highly accessible to diverse research groups. I have a number of suggestions below. Given this is a short report they can likely all be addressed with minor text edits. Overall, this work was a pleasure to read and opens up exciting new avenues of investigation for multiple fields of nutrition and neuroscience. 

1. The data in 1L may be improved by quantification, or at the very least higher magnification images.

2. The Dilp5 results seem to be overinterpreted. The reduced levels could also be due to reduced synthesis under the conditions, and this needs to be documented at some form (at least at the transcript level).

3. The manuscript does not address the specific effects of high sugar diet versus increased caloric intake. Including this in the discussion, or experimentally, would increase impact. A few sentences on this matter would add to the impact of the discussion.

4. While discussed in tangential ways, please include a direct comparison of the similarities and differences between ensheathing glia and microglia. In addition, the discussion does not include whether the effects observed are specific to microglia or generalizable to all glial types. 

5. I have some concern about the use of maximum intensity to quantify images. I recognize that this is a standard in the field, however, it masks a significant amount of data throughout the Z-stack. Perhaps a supplemental movie depicting the full Z-stack for critical data sets would increase data accessibility.

Reviewer #2 Girish C. Melkani, UAB Heersink of Medicine (note, reviewer 2 has signed this review): In this short communication, using a Drosophila model, the author shows that a chronic obesogenic diet induces glial insulin resistance and impairs the clearance of neuronal debris by downregulating the glia-associated phagocytic receptor, Draper. Genetically attenuating systemic insulin release rescues diet-induced glial insulin resistance and draper expression, while stimulating Phosphoinositide 3-kinase (PI3K) rescues high-sugar diet-induced glial defects. This study establishes that obesogenic diets impair glial phagocytic function, resulting in delayed clearance of neuronal debris. This is an interesting finding, and we need more studies like this, however, some additional assays will be required to make these findings complete and more impactful. Therefore, the following questions need to be addressed for consideration in PLoS Biology

1. Many downstream targets of insulin receptors are known, including Stat92E, however, it is unclear why PI3K was selected. The author should confirm the expression of other downstream targets with qPCR. 

2. Other complications linked with obesity have not been discussed have nor other obesogenic conditions. 

3. Obesogenic diet is not limited to HSD, Does the author report a similar phenotype with HFD and/or missed diets (HSD+HFD), which will mimic with human obesogenic diet?

4. The author has indicated that microglia are the brain's resident macrophages, and they play a key role in learning and memory. However, no attempt is made to test memory impairment linked with HSD and other genetic modulations the authors have made this question. It is unclear what makers are used for neuroinflammation.

5. I am unclear how the authors rule out the first source of energy balance, i.e., glycolysis in Fig1. Multiple findings showed that this can also be a source of energy balance. 

6. Mitochondrial morphology alteration in the fat tissue in revealing in the brain function is not convincing for mitochondrial function, as their shape is unchanged in the brain (fig. 1 E, F). Is there interorgan communication from fat tissue to the brain? If yes, this must be addressed. 

7. Fig. 1K is convincing, however, quantification will be required, which is lacking. 

8. It is unclear in Figs. 2-4, what basis rectangles were selected for data analyses? Not clarified what area of the brain is used for quantification. It seems biased approach, and it must clarify, including a detailed description in the method section. 

9. Overall, this interesting finding is limited to cytological/genetic analyses, and these data along with the behavior/functional analyses will make this more impactful and will be suitable for publication in PLoS Biology. 

Additional comments:

1. Upon chronic exposure to an HSD, circulating insulin levels rise, resulting in insulin resistance---------------------------it should be circulating sugar levels rise, correct.

2. Insulin resistance linked with obesogenic challenges has been tested in Drosophila peripheral organs, however, this was not discussed. 

3. The figure legends and method sections are brief and do not provide sufficient information. These should be addressed in the revised version. 

Reviewer #3: The manuscript by Alassaf and Rajan elegantly reveals that insulin resistance is induced in the adult Drosophila brain by a chronic high sugar diet (HSD). The authors nicely show that this diet delays clearance of injured axon debris by ensheathing glia as a result of reduced levels of the phagocytic receptor Draper (basal and injury-induced). 

This interesting and very well written manuscript exposes a possible connection between obesogenic diets and impaired clearance of neuronal debris in adult brain, which might be implicated in understanding the pathophysiology of neurodegenerative diseases. The findings are novel and significant for a wide research community. 

The authors focus on the specific type of glia, ensheathing glia, which was shown to perform phagocytosis of axonal debris. Since neurodegeneration can involve different mechanisms from injury-induced wound healing mechanism, other types of glia might be also involved. Cortex and astrocyte-like glia, which can be also phagocytic were previously shown to response differently to Insulin signaling ("Independent glial subtypes delay development and extend healthy lifespan upon reduced insulin-PI3K signaling" by Nathaniel S. Woodling et al., 2020). It is interesting to know whether HSD-induced or systemic insulin resistance in other types of glia also affect Draper expression in these cells. 

In Figure 1L statistics of accumulation of lipid droplets is missing. Do other types of glia also accumulate lipid droplets? In some cases, accumulation of lipid droplets can be neuroprotective ("Antioxidant Role for Lipid Droplets in a Stem Cell Niche of Drosophila" by Andrew P Bailey et al., 2015). It is worth to discuss it.

What happens to mitochondria in neurons? Do they look differently on HSD and under normal conditions?

Minor comments:

- All gene names and genotypes must be in italics everywhere including Figures and Fly husbandry, no capital letters (page 5 "draper gene"), protein names must start with a Capital letter (Draper - line 13 in the abstract)

- LipidTOX is written differently in different places

- Please remove pointing to Figures from Discussion

- In "image analysis" there is "A" in the middle of the sentence: "To measure mitochondrial morphology, a maximum…"

---

## [Decision Letter · Decision Letter 2]

22 Sep 2023

Dear Dr Rajan,

Thank you for your patience while we considered your revised manuscript "Diet-induced glial insulin resistance impairs the clearance of neuronal debris." for publication as a Short Report at PLOS Biology. This revised version of your manuscript has been evaluated by the PLOS Biology editors, the Academic Editor and the original reviewers, who are fully satisfied by the revision and have no additional requests. 

While we are likely to accept this manuscript for publication, based on this feedback, before we can formally accept your study, we need you to address a number of minor editorial requests in another revision that we think will not take very long. 

**Please address the following editorial requests: 

1) FINANCIAL DISCLOSURES: Please update your financial disclosures statement, in the relevant section of our online system, to describe the role of any sponsors or funders in the study design, data collection and analysis, decision to publish, or preparation of the manuscript. If the funders had no role in any of the above, include this sentence at the end of your statement: "The funders had no role in study design, data collection and analysis, decision to publish, or preparation of the manuscript.

2) BLURB: In the relevant section of our online system, please provide a blurb which (if accepted) will be included in our weekly and monthly Electronic Table of Contents, sent out to readers of PLOS Biology, and may be used to promote your article in social media. The blurb should be about 30-40 words long and is subject to editorial changes. It should, without exaggeration, entice people to read your manuscript. It should not be redundant with the title and should not contain acronyms or abbreviations.

3) DATA: Thank you for providing the data underlying your figures as supplemental excel files. I noticed that the data for Fig 1D seemed to be missing. Can you please provide this? Please also be sure to add a brief sentence to each figure legend (including supplemental) referencing the relevant supplemental file where the underlying data can be found. 

4) CODE: Per journal policy, if any code was generated to support the conclusions of your manuscript, we require that you make it available without restrictions upon publication. Please ensure that any code is sufficiently well documented and reusable, and that your Data Statement in the Editorial Manager submission system accurately describes where your code can be found.

We expect to receive your revised manuscript within two weeks. 

*Published Peer Review History*

*Press*

Sincerely,

Luke

Lucas Smith, Ph.D.

Senior Editor,

lsmith@plos.org,

PLOS Biology

Reviewer remarks:

Reviewer #1, Alex C. Keene (note, reviewer 1 has signed this review): In this revised manuscript the authors have fully addressed my concerns. The conclusions are now supported by additional analyses, and text change provide clarity with respect to the link between ensheathing and microglia, and the obesigenic diet used. I have no additional concerns and appreciate the authors' responsiveness.

Reviewer #2: The author is really wonderful job to address the questions/comments in the previous version. I do not have any reservations about the findings of this study should be suitable for publication in PLoS Biology. 

Reviewer #3: The authors addressed all my concerns in a good faith.

---

## [Editor Report · Decision Letter 3]

3 Oct 2023

Dear Akhila,

Thank you for the submission of your revised Short Report, "Diet-induced glial insulin resistance impairs the clearance of neuronal debris", for publication in PLOS Biology, and thank you for addressing our last editorial requests in this revision. On behalf of my colleagues and the Academic Editor, Mikael Simons, I am pleased to say that we can in principle accept your manuscript for publication, provided you address any remaining formatting and reporting issues. These will be detailed in an email you should receive within 2-3 business days from our colleagues in the journal operations team; no action is required from you until then. Please note that we will not be able to formally accept your manuscript and schedule it for publication until you have completed any requested changes.

**IMPORTANT: As you address any formatting and reporting requests to come, we also ask that you address the following minor editorial request, which I think was missed in the last revision. 

1) Please add a brief sentence to each figure legend (including supplemental) referencing the relevant supplemental file where the underlying data can be found. For example, to each figure legend, you can add the sentence "The data underlying this figure can be found in ____" (and then reference the relevant excel file). 

PRESS

Sincerely, 

Luke

Lucas Smith, Ph.D.

Senior Editor

PLOS Biology

lsmith@plos.org